# Multi-trait discovery and fine-mapping of lipid loci in 125,000 individuals of African ancestry

Abram Bunya Kamiza [1,2,3,19], Sounkou M. Touré [1,4,19], Feng Zhou [5,19], Opeyemi Soremekun[1], Cheickna Cissé[4,6], Mamadou Wélé[2,6], Aboubacrine M. Touré[6], Oyekanmi Nashiru[7], Manuel Corpas [8], Moffat Nyirenda[9], Amelia Crampin[2], Jeffrey Shaffer[10], Seydou Doumbia[4,11], Eleftheria Zeggini [12,13], Andrew P. Morris [14], Jennifer L. Asimit [5], Tinashe Chikowore [3,15,16,17] & Segun Fatumo [1,7,12,18] ✉

Most genome-wide association studies (GWAS) for lipid traits focus on the separate analysis of lipid traits. Moreover, there are limited GWASs evaluating the genetic variants associated with multiple lipid traits in African ancestry. To further identify and localize loci with pleiotropic effects on lipid traits, we conducted a genome-wide meta-analysis, multi-trait analysis of GWAS (MTAG), and multi-trait fine-mapping (flashfm) in 125,000 individuals of African ancestry. Our meta-analysis and MTAG identified four and 14 novel loci associated with lipid traits, respectively. flashfm yielded an 18% mean reduction in the 99% credible set size compared to single-trait fine-mapping with JAM. Moreover, we identified more genetic variants with a posterior probability of causality >0.9 with flashfm than with JAM. In conclusion, we identified additional novel loci associated with lipid traits, and flashfm reduced the 99% credible set size to identify causal genetic variants associated with multiple lipid traits in African ancestry.

Lipid traits including high-density lipoprotein cholesterol (HDL-C), low-density lipoprotein cholesterol (LDL-C), triglycerides (TG) and total cholesterol (TC) are implicated in cardiometabolic diseases[1,2]. Evidence from epidemiological studies indicates that cardiometabolic disease rates in Africa are comparable to those in other parts of the world, but are rapidly increasing due to unhealthy dietary intake patterns and lifestyle factors[3,4]. Moreover, previous studies have suggested that individuals from continental Africa are associated with less atherogenic lipid profiles[5]. However, individuals of African ancestry living in the US or Europe have higher rates of cardiometabolic diseases than other ancestry groups[5–7], suggesting that ancestral differences exist in the aetiology of cardiometabolic diseases.

Lipid traits are influenced by environmental and genetic factors[8]. Unhealthy dietary intake, physical inactivity, cigarette smoking, alcohol consumption and several genetic factors are some of the factors implicated in atherogenic lipid levels. Although several genetic loci are associated with lipid traits[9–11], they explain only a small fraction of the variances of atherogenic lipid levels[9]. Moreover, these genetic loci were mainly discovered in European and East Asian ancestry cohorts[12–14], and the transferability of these genetic loci to individuals of African ancestry has been poor[15], probably due to differences in linkage disequilibrium (LD), allele frequencies and environmental exposure among other factors, suggesting that more genetic variants remain to be discovered. To address this, The Global Lipids Genetics Consortium (GLGC) performed a meta-analysis of genome-wide association studies (GWASs) with more than 1.6 million individuals from five ancestries. Of these individuals, 99,432 were of African ancestry, in whom an additional 15 novel loci associated with lipid traits were identified[16].

Although the GLGC meta-analysis included 99,432 individuals of African ancestry, 72,859 were African Americans in the US, which

may not carry ancestry-specific genetic variants across Africa. Moreover, the majority of African Americans in the US carry West African ancestry. To further identify genetic loci associated with lipid traits in individuals of African ancestry and determine their molecular mechanisms, putative causal genetic variants, and cover greater genetic diversity across the continent of Africa, we performed a meta-analysis of GWAS including up to 125,000 individuals of African ancestry. Of these individuals ~14,000 were from the African Partnership for Chronic Disease Research (APCDR) consortium in Africa, ~99,000 were from the GLGC and ~11,000 were from Africa Wits-INDEPTH Partnership for Genomic Research (AWI-Gen) in Africa. To increase the statistical power of detecting additional novel genetic loci associated with lipid traits, we used the multi-trait analysis of GWAS (MTAG)[17] approach by taking advantage of the correlation among the lipid traits. Analyzing multiple lipid traits simultaneously can provide more accurate and robust results than analyzing each trait separately. Moreover, we used a single trait (JAM)[18] and multi-trait (flashfm)[19] fine-mapping methods to identify causal genetic variants associated with lipid traits in individuals of African ancestry; sharing information between traits by joint fine-mapping with flashfm results in higher resolution (smaller credible sets) than fine-mapping each trait independently.

## Results

### Meta-analysis of GWAS

We conducted a GWAS meta-analysis of ~125,000 individuals of African ancestry and identified 63, 68, 48, and 92 independent genetic loci (500 kb around lead SNP) associated with HDL, LDL, TG and TC, respectively, at genome-wide significance ($p$-value $< 5 \times 10^{-8}$) (Supplementary Data 1). The Manhattan and quantile-quantile (QQ) plots for all lipid traits are shown in Fig. 1. Of the independent genetic loci, four were novel and not previously reported to be associated with lipid traits (Table 1). The variants rs2451303 ($p$-value $= 8.0 \times 10^{-09}$) and rs6704760 ($p$-value $= 4.50 \times 10^{-08}$) were associated with LDL and mapped to the intergenic regions between *NT5IB* and *RDH14*, *MYCN* and *MYCNOS*, respectively, Fig. 2. The variants rs12118522 ($p$-value $= 4.52 \times 10^{-08}$) and rs115505361 ($p$-value $= 4.49 \times 10^{-08}$) were associated with TG, mapped within *CHRM3* and *MGAT2*, and associated with hypertension, body fat distribution and body mass index and blood protein levels, respectively, Fig. 2. To determine the similarities or differences across the three datasets used in the meta-analysis, we identified the top hits in GLGC-AFR compared to those in APCDR and AWI-Gen (Supplementary Fig. 1). We found that these loci were consistent across the three datasets, although some were not genome-wide significant in the APCDR and AWI-Gen datasets.

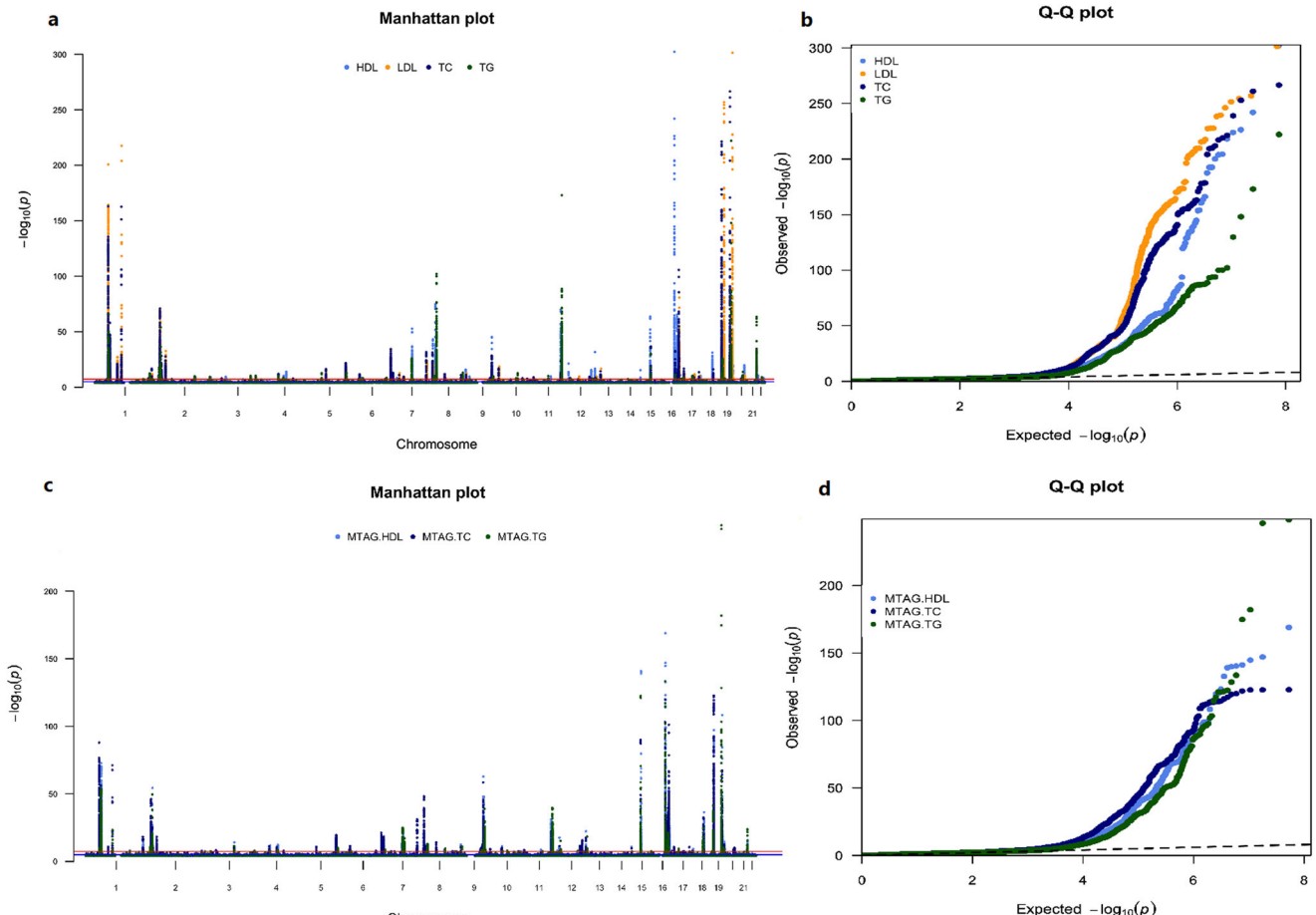

**Fig. 1 | Genome-wide association study (GWAS) for lipid traits in individuals of African ancestry. a**, **b** are Manhattan and QQ plot, respectively for GWAS meta-analysis ($N = ~125,000$). **c, d** are Manhattan and QQ plot, respectively for multi-trait analysis of GWAS (MTAG, $N = ~125,000$). X-axis are the genomic position, the Y-axis represents the log10 of association $p$-values and the point above the dotted line represented variants significant at $p$-value $< 5 \times 10^{-8}$. $P$-values are two-tailed calculated using GWAMA for meta-analysis and MTAG for multi-trait analysis of GWAS, not adjusted for multiple comparisons. Of the 87 (seven novel) loci for the meta-analysis of GWAS and 85 (13 novel) loci for MTAG analyses corresponding to 107 distinct loci (65 shared) were identified. In **b**, and **d** is a comparison of the expected distribution of association −log10 $p$-values under the null distribution. Leftward deviation of the curves from the dotted line of expected values indicated more associated loci than expected.

**Table 1 | Novel genetic loci associated with lipid traits in individuals of African ancestry from meta-analysis genome-wide association study**

| CHR | SNP | BP | A1 | A2 | EAF | BETA | SE | P-value | N | Genes | Trait |
|---|---|---|---|---|---|---|---|---|---|---|---|
| 1 | rs12118522 | 239643848 | G | T | 0.038 | −0.068 | 0.012 | 4.49 x 10$^{-08}$ | 107148 | CHRM3 | TG |
| 2 | rs6704760 | 16069674 | A | T | 0.096 | 0.043 | 0.007 | 7.97 x 10$^{-09}$ | 105011 | MYCNUN;MYCNOS | LDL |
| 2 | rs2451303 | 18642470 | G | A | 0.318 | 0.027 | 0.005 | 4.50 x 10$^{-08}$ | 105010 | NT5C1B – RDH14 | LDL |
| 14 | rs115505361 | 49510761 | G | A | 0.032 | 0.068 | 0.012 | 4.52 x 10$^{-08}$ | 118814 | MGAT2 | TG |

*CHR* chromosome, *SNP* single nucleotide polymorphism, *BP* base position, *A1* effect allele, *A2* other allele, *EAF* effect allele frequency, *SE* standard error, *N* sample size.
*P*-values are two-tailed calculated using GWAMA and not adjusted for multiple comparisons.

## MTAG identified additional novel loci

We then proceeded to perform a multi-trait analysis of GWAS (MTAG)[17] to increase the statistical power to identify additional shared genetic variants associated with lipid traits. We applied the fixed-effect meta-analysis with linkage disequilibrium scores (LDSC) estimated from the 1000 Genome Project phase 3 of individuals of African ancestry[20] (Methods). Our MTAG approach, identified 84, 62, and 110 independent genetic loci associated with HDL, TG, and TC, respectively, at genome-wide significance (*p*-value < 5 × 10$^{-8}$, 500 kb) (Supplementary Data 2). The Manhattan and QQ plots for HDL, TG, and TC are shown in Fig. 1. Of the independent genetic loci, 14 were novel and not previously reported to be associated with HDL, TG, and TC (Table 2). The strongest novel loci associated with lipid traits for our MTAG approach were mapped within *TMEM64* (rs74979471, *p*-value < 4.75 x 10$^{-10}$), *ZNF782* (rs6477710, *p*-value < 1.36 x 10$^{-10}$), intergenic region between *MSANTD3* and *TMEFF1* (rs113951466, *p*-value < 6.01 x 10$^{-10}$), and *LOC100507346* (rs7850215, *p*-value < 2.06 x 10$^{-09}$, Supplementary Fig. 2). Of the 14 novel genetic loci associated with HDL, TG, and TC for the MTAG approach, five (rs368884688, rs6477710, rs113951466, rs7850215, and rs10819792) overlap among the lipid traits (Table 2). Of the novel loci associated with lipid traits in individuals of African ancestry, four and eight did not exist in the GLGC summary data of European and East Asians ancestry, and those that were present were not significantly associated (*p*-value = 0.0124) with lipid traits (Supplementary Data 3). These findings highlight the importance of studying individuals of African ancestry in the context of GWAS. We then compared the number of independent genomic loci identified by our meta-analysis and MTAG approaches (Supplementary Fig. 3). Notably, our MTAG increased the discovery of genetic loci by 25%, 25% and 17.3% for HDL, TG and TC, respectively compared to univariate meta-analysis.

## Fine mapping of associated loci

We then performed fine mapping to localize putative causal variants associated with lipid traits by taking advantage of the small linkage disequilibrium block structure among individuals of African ancestry. Fine-mapping was performed using JAM[18], a single-trait fine-mapping method and flashfm[19], a multi-trait fine-mapping method, using the AFR super population of 1000 Genomes and an in-sample subset of individuals of African ancestry as a reference panel (Methods)[20].

The lipid trait correlations between LDL, TC, TG, and HDL indicate a high correlation between LDL and TC, and a low to moderate correlation between all other trait pairs (Fig. 3a). In comparing 99% credible sets (CS99) between JAM and flashfm, we found that flashfm gave a 17.6% mean reduction in CS99 size over JAM. Moreover, 93% (114/122) of the CS99 were either the same size or refined by flashfm, and 60.6% (74/122) of the CS99 are strictly smaller than those from JAM (Fig. 3b).

In 15% of the flashfm CS99, there was at least one variant with a marginal posterior probability (MPP) of being a causal variant (MPP) > 0.90, whereas all JAM variants had MPP < 0.9 (Supplementary Data 4). We highlight three regions in Table 3, where flashfm refines the CS99 over JAM for a trait(s) and flashfm results in at least one variant with MPP > 0.9, though JAM does not. In particular, flashfm and JAM agree on the most likely causal SNP(s) for some traits and give similar MPP, but there are other traits for which; (1) flashfm assigns noticeably higher MPP than JAM for the same variant (i.e. rs78302875 increases from 0.406 (JAM) to 0.920 (flashfm) for HDL in 16:71465787-71665787), (2) flashfm assigns noticeably higher MPP than JAM for a variant in high LD with it (i.e.. the max MPP in JAM is 0.501 for rs247616, which has $r2 = 0.992$ with rs183130, the top SNP for flashfm (MPP = 0.999) for LDL in 16:56889590-57089590) (Supplementary Fig. 4), (3) flashfm assigns noticeably higher MPP than JAM for a variant in moderate LD with it (i.e. the max MPP in JAM is 0.332 for rs4783961, which has $r2 = 0.470$ with rs183130, the top SNP for flashfm (MPP = 0.999) for TG in 16:56889590-57089590) (Supplementary Fig. 4) and (4) flashfm gives an 87.5% reduction from 16 variants in the JAM CS99 to 2 variants

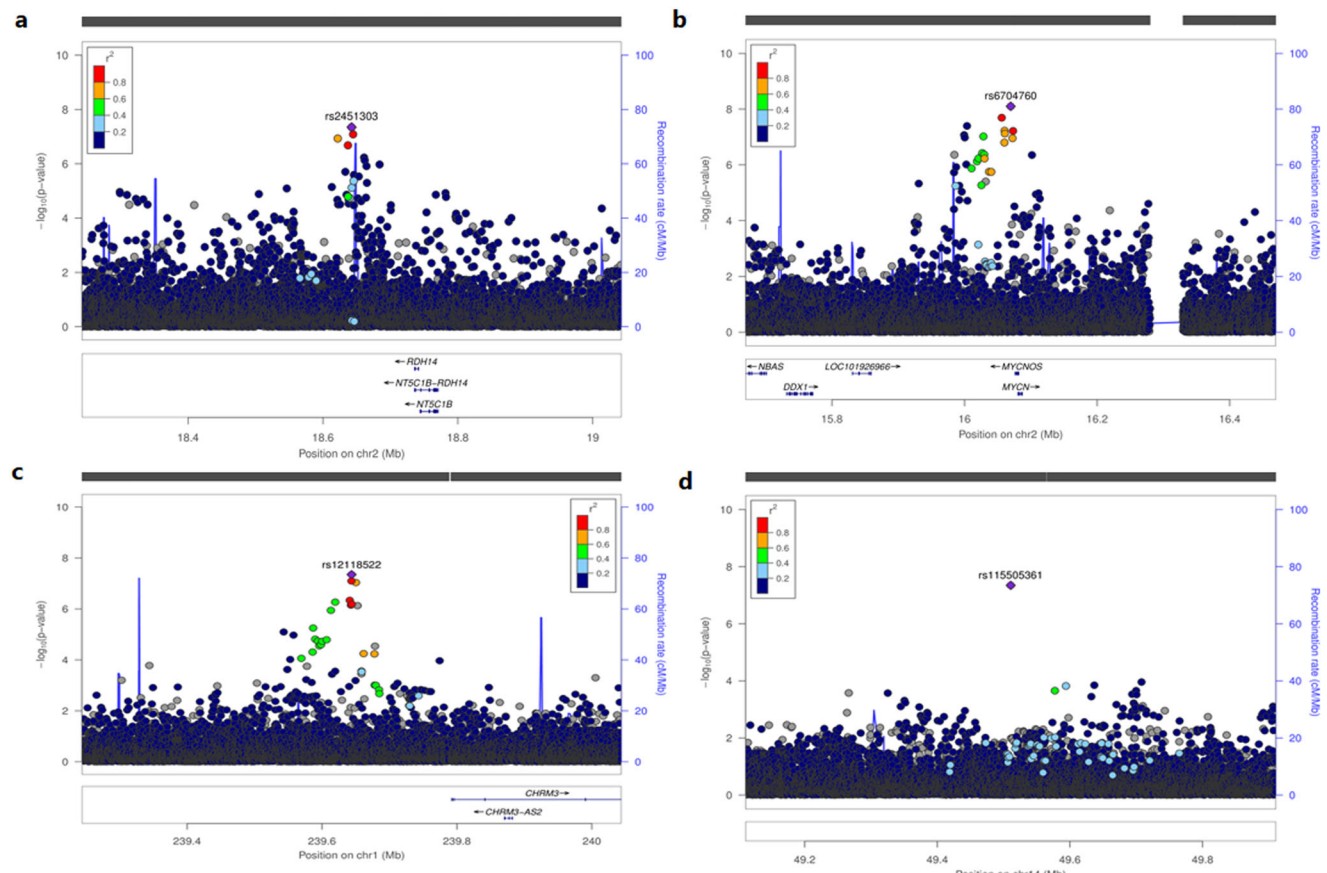

**Fig. 2 | Novel loci associated with lipid traits in individuals of African ancestry from the meta-analysis genome-wide association analysis (N = ~ 125,000).** **a** Locuszoom plot showing associations around the intergenic regions between *NT5IB* and *RDH14* region. **b** Locuszoom plot showing associations around the intergenic region between *MYCN* and *MYCNOS*. **c** Locuszoom plot showing associations around the *CHRM3* region. **d** Locuszoom plot showing associations around the *MGAT2* region.

**Table 2 | Novel genetic loci associated with lipid traits in individuals of African ancestry from multi-trait genome-wide association study**

| CHR | SNP | BP | A1 | A2 | EAF | BETA | SE | P-value | N | Genes | Traits |
|---|---|---|---|---|---|---|---|---|---|---|---|
| 1 | rs368884688 | 181315526 | G | A | 0.994 | −0.267 | 0.046 | 7.20 × 10⁻⁰⁹ | 13114 | *CACNA1E* | HDL |
| 1 | rs368884688 | 181315526 | G | A | 0.994 | −0.277 | 0.050 | 3.47 × 10⁻⁰⁸ | 13115 | *CACNA1E* | TG |
| 3 | rs149700703 | 45204966 | T | G | 0.008 | 0.232 | 0.040 | 5.68 × 10⁻⁰⁹ | 31019 | *RPS24P8* | TC |
| 4 | rs911749026 | 6362109 | A | G | 0.991 | −0.229 | 0.041 | 1.64 × 10⁻⁰⁸ | 13115 | *PPP2R2C* | TG |
| 7 | rs13438114 | 31039982 | A | G | 0.189 | 0.025 | 0.005 | 4.85 × 10⁻⁰⁸ | 123032 | *GHRHR* | TC |
| 8 | rs77237080 | 91701696 | G | A | 0.064 | −0.028 | 0.005 | 3.59 × 10⁻⁰⁹ | 120770 | *TMEM64* | HDL |
| 8 | rs74979471 | 91724679 | A | G | 0.070 | −0.031 | 0.005 | 4.75 × 10⁻¹⁰ | 119930 | *TMEM64* | TG |
| 8 | rs112773301 | 133214483 | G | A | 0.986 | 0.134 | 0.024 | 4.65 × 10⁻⁰⁸ | 23589 | *KCNQ3* | TG |
| 9 | rs6477710 | 99585314 | G | T | 0.069 | −0.032 | 0.005 | 1.36 × 10⁻¹⁰ | 102013 | *ZNF782* | HDL |
| 9 | rs113951466 | 103289530 | T | C | 0.077 | −0.027 | 0.004 | 6.01 × 10⁻¹⁰ | 120770 | *MSANTD3-TMEFF1:TMEFF1* | HDL |
| 9 | rs7850215 | 98327711 | A | G | 0.104 | −0.023 | 0.004 | 2.06 × 10⁻⁰⁹ | 120771 | *RP11-332M4.1* | HDL |
| 9 | rs6477710 | 99585314 | G | T | 0.070 | −0.047 | 0.008 | 5.65 × 10⁻¹⁰ | 104276 | *ZNF782* | TC |
| 9 | rs4743855 | 94581258 | T | C | 0.107 | −0.024 | 0.004 | 2.08 × 10⁻⁰⁹ | 110284 | *ROR2* | HDL |
| 9 | rs10819792 | 98834653 | T | C | 0.219 | −0.016 | 0.003 | 6.03 × 10⁻⁰⁹ | 120771 | *RP11-569G13.2* | HDL |
| 9 | rs7041800 | 95951769 | A | G | 0.072 | −0.043 | 0.007 | 4.72 × 10⁻⁰⁹ | 104276 | *WNK2* | TC |
| 9 | rs10819792 | 98834653 | T | C | 0.220 | −0.024 | 0.004 | 1.36 × 10⁻⁰⁸ | 123033 | *RP11-569G13.2* | TC |
| 9 | rs113951466 | 103289530 | T | C | 0.077 | −0.027 | 0.005 | 9.84 × 10⁻⁰⁹ | 119930 | *MSANTD3-TMEFF1:TMEFF1* | TG |
| 9 | rs7850215 | 98327711 | A | G | 0.105 | −0.033 | 0.006 | 1.68 × 10⁻⁰⁸ | 123032 | *RP11-332M4.1* | TC |
| 12 | rs374366861 | 87353564 | T | C | 0.943 | 0.070 | 0.012 | 1.87 × 10⁻⁰⁸ | 23589 | *RP11-324H9.1* | TG |

*CHR* chromosome, *SNP* single nucleotide polymorphism, *BP* base position, *A1* effect allele, *A2* other allele, *EAF* effect allele frequency, *SE* standard error, *N* sample size.
*P*-values are two-tailed calculated using MTAG and not adjusted for multiple comparisons.

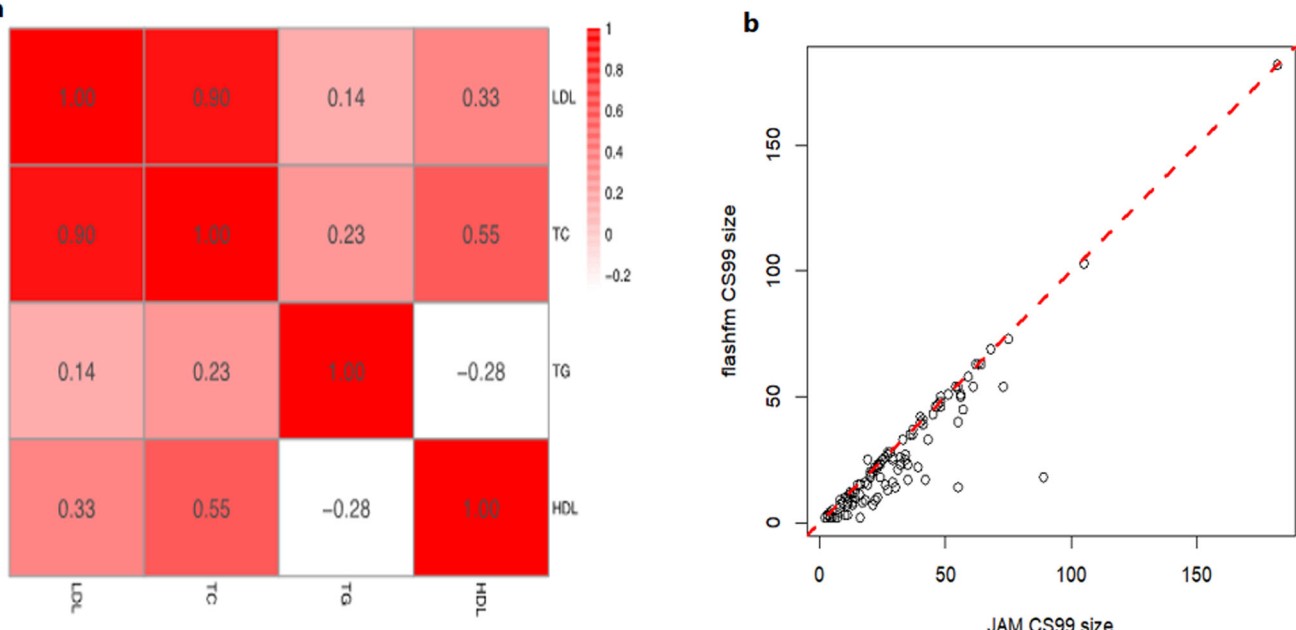

**Fig. 3 | Fine-mapping of lipid traits in individuals of African ancestry. a** Lipid trait correlation measured as Pearson's correlation in the Uganda genome resource ($N$ = ~125, 000). **b** flashfm multi-trait fine-mapping generated smaller 99% credible sets (CS99) than JAM single-trait fine-mapping ($N$ = ~125,000).

in the flashfm CS99 for TC and 21:46753876-46975775, and the variant favoured by JAM, rs77974343 (MPP = 0.275) has moderate LD ($r2$ = 0.686) with the variant favoured by flashfm, rs116386571 MPP = 0.989) (Supplementary Fig. 5). The variants rs77974343 and rs116386571 are both intronic low-frequency variants (African MAF = 0.02; European MAF = 0) and have been previously identified in African ancestries as associated with TG (rs77974343, p-value = 5.77 x 10$^{-47}$; rs11638657, p-value = 6.03 x 10$^{-49}$) and TC (rs77974343, p-value = 7.61 x 10$^{-10}$; rs11638657, p-value = 1.14 x 10$^{-6}$)[16]. HaploReg[21] indicates that rs116386571 has enhancer histone marks for adipose nuclei and fetal heart, as well as bound proteins *TCF12* and *POL2B* and 18 altered motifs, whereas rs77974343 has three altered motifs and enhancer histone marks in the spleen, suggesting slightly more biological support for rs116386571.

### Comparison of fine-mapping results

We compare our African ancestry JAM and flashfm results with those of GLGC: single causal variant single trait fine-mapping for admixed African/African ancestry and trans-ancestry[16]. We compare the lists of variants with MPP > 0.90 for each method, for instances where our region contains the lead SNP of the region considered by GLGC. This resulted in 90 of our trait-region combinations that overlap with GLGC, of which 62 have flashfm variants with MPP > 0.90 for comparison (Supplementary Data 4). Among these 62 regions, 31 have variants with PP > 0.90 in the GLGC African analysis, and 48 have variants with PP > 0.90 in the GLGC trans-ancestry analysis; we compare our results with those of GLGC within these regions where both flashfm and one of the GLGC analyses has at least one variant with MPP > 0.90.

As expected, there is a higher agreement between our analysis with the African analysis of GLGC, than with their trans-ancestry analysis. In 45% (14/31) of the regions that have variants prioritised (MPP > 0.90) by both flashfm and the African fine-mapping of GLGC, there was a shared prioritised variant; 21% (10/48) of the regions that have variants prioritised by both flashfm and the multi-ancestry fine-mapping of GLGC share a prioritised variant. Details of variants prioritised by flashfm or JAM, as well as all functional annotations and whether they are also prioritised by the GLGC analyses are also provided (Supplementary Data 5).

### MTAG identifies eQTL with a high posterior probability of shared association

To identify putative functional mechanisms of novel loci, we performed Bayesian colocalization on meta-analysis and MTAG summary data with GTEx v8 tissue-specific gene expression quantitative trait loci (eQTL) using coloc[22]. Of the four novel loci identified by meta-analysis, none were colocalized (H4 PP < 0.8) with the respective lipid traits (Supplementary Data 6). For the MTAG summary data, we found six genetic loci (rs13438114, rs6477710, rs113951466, rs4743855, rs10819792, and rs7041800) with H4 PP > 0.8; the co-localized gene expression traits involved multiple tissues (Supplementary Data 7). For instance, *ADCYAP1R1, ANKRD19P, CCDC180, ECM2, FGD3, HABP4, LINC02937, MSANTD3, NUTM2G, OGN, IPPK, PTLC1*, and *ZNF484* had H4 PP > 85% with MTAG HDL and were expressed in several tissues, notably those of the liver, brain, adipose tissues, artery, gastrointestinal tract, and whole blood (Supplementary Data 7). Interestingly, these genes are involved in several pathways, including insulin secretion, renin secretion, the cAMP signaling pathway, inositol phosphate metabolism, metabolic pathways, and the phosphatidylinositol signaling system.

## Discussion

By analyzing multiple lipid traits simultaneously, we have provided more accurate and robust results than by analyzing each lipid trait separately. Moreover, we replicated several genetic loci previously reported to be associated with lipid traits including, *PSCK9, APOE, LPL, CETP*, and *DOCK7* in individuals of African ancestry[22]. The greater coverage of African genetic variation allowed us to identify an additional four novel genetic loci in our GWAS meta-analysis. Using the MTAG approach, we identified 14 additional novel genetic loci associated with lipid traits that were not found in our GWAS meta-analysis approach. Moreover, our MTAG gene expression trait colocalization found six loci with a posterior probability of shared causality ≥80%. For fine mapping, we observed an improvement in terms of credible set size reduction when multiple lipid traits were jointly fine-mapped using flashfm.

Our meta-analysis of AWI-Gen, APCDR, and GLGC-AFR, found four novel loci associated with lipid traits in individuals of African ancestry (Table 1). The rs115505361 is mapped on *MGAT2* and associated with

**Table 3 | Comparison of fine-mapping results between JAM single trait and flashfm multi-trait fine mapping**

| Region (Nearest genes) | Trait | JAM CS99 size | flashfm CS99 size | Size Reduction (%) | JAM top SNPs | flashfm top SNPs | JAM MPP of top SNPs | flashfm MPP of Top SNPs | LD (r²) between JAM & flashfm top SNPs |
|---|---|---|---|---|---|---|---|---|---|
| 16:56889590-57089590(CETP, HERPUD1) | HDL | 24 | 24 | 0 | >10 | >10 | all >0.9 | all >0.9 | 1 |
| | LDL | 5 | 2 | 60 | rs247616 | rs183130 | 0.501 | 0.999 | 0.992 |
| | TC | 24 | 23 | 4.2 | rs1801706 | rs1801706 | 0.925 | 0.983 | 1 |
| | TG | 7 | 2 | 71.4 | rs4783961 | rs183130 | 0.332 | 0.999 | 0.470 |
| 16:7146587-7166587(CHSY4, ZNF19, TAT) | HDL | 8 | 4 | 50 | rs78302875 | rs78302875 | 0.406 | 0.920 | 1 |
| | LDL | 43 | 33 | 23.3 | rs77223082 rs144454811 rs72799815 rs72799820 | rs77223082 rs144454811 rs72799815 rs72799820 | all >0.9 | all >0.9 | 1 |
| | TC | 21 | 20 | 4.8 | rs115532837 rs57422486 rs78302875 rs144454811 rs116408331 rs72799815 rs72799820 | rs115532837 rs57422486 rs78302875 rs144454811 rs116408331 rs72799815 rs72799820 | all >0.9 | all >0.9 | 1 |
| 21:46753876-46975775(SLC19A1, COL18A1) | HDL | 18 | 16 | 11.1 | rs116386571 | rs116386571 | 0.863 | 0.999 | 1 |
| | TC | 16 | 2 | 87.5 | rs77974343 | rs116386571 | 0.275 | 0.989 | 0.686 |
| | TG | 17 | 8 | 52.9 | rs116386571 | rs116386571 | 1.0 | 1.0 | 1 |

CS99 size; 99 credible set size, MPP marginal posterior probability of causality.

body mass index, blood proteins, insulin sensitivity and hypertriglyceridemia. *MGTA2* is involved in various metabolic and N-glycan biosynthesis pathways, including lipid metabolism and protein synthesis. rs12118522, mapped to *CHRM3*, is associated with hypertension and body fat distribution. Some pathways associated with *CHRM3* include calcium signalling, cholinergic synapses, taste transduction, insulin secretion, salivary secretion, gastric acid secretion, and pancreatic secretion pathways. rs2451303 and rs6704760 are novel loci associated with lipid traits and are mapped in the intergenic regions between *NT51B* and *RDH14*, *MYCN* and *MYCNOS*, respectively. These genes have been previously reported to be associated with cardiovascular diseases and neoplasm, respectively.

As hypothesized, by using a multivariate approach, we improved the discovery of additional novel loci (Table 2). The MTAG approach identified 84, 62, and 110 distinct genetic loci associated with HDL, TG, and TC levels, respectively. Moreover, on average, MTAG analysis increased the power of identifying additional loci by 25% and provided greater power for downstream pathway analysis (Fig. 3). We also noted that the MTAG approach produced more novel genetic loci than GWAS meta-analysis results.

Notably, our novel genetic loci have been linked to traits correlated with blood lipid traits, including body mass index, systolic and diastolic blood pressure, diabetes, chronic kidney disease, blood protein levels, blood and immunology phenotypes, and Alzheimer's biomarkers such as amyloid or infectious diseases. For instance, *CACNA1E* encodes the alpha-1E subunit of R-type calcium channels, which belong to the 'high-voltage activated' group that may be involved in the modulation of firing patterns of neurons important for information processing[23]. This gene is involved in the following pathways; MAPK signaling, calcium, T2D, and early infantile epileptic encephalopathy pathways and has been previously associated with blood pressure, BMI, and coronary artery calcification. *TMEM64* is localized in the endoplasmic reticulum and modulates the nuclear localization of β-catenin, resulting in the activation of β-catenin-mediated transcription[24]. Diseases and phenotypes associated with *TMEM64* include cholesterol, creatinine, chronotype, and serum metabolite concentrations in chronic kidney disease. *PPP2R2C* has previously been associated with BMI, coronary artery disease, T2D, and metabolite levels. This gene belongs to the phosphatase 2 regulatory subunit B family[25], which is related to PI3K-Akt, AMPK signaling, adrenergic signaling in cardiomyocytes, Chagas disease, hepatitis C, and human papillomavirus infection pathways. These pathways suggest an important role of viral infections in lipid regulation in areas with a high viral load. This finding is consistent with the impact of viral infection on lipid metabolism. Indeed, gene and pathway analyses confirmed that inflammatory pathways and beta-cell type gene sets are associated with these traits. Future GWAS with imputed HLA variants may reveal additional insights into the relationship between lipid regulation and immunity, which has been observed in a previous study[26].

Our MTAG gene expression trait colocalization analysis identified six loci with an H4 PP ≥ 80%. These genes are involved in several pathways, including insulin secretion, renin secretion, cAMP signaling, inositol phosphate metabolism, metabolic pathways, and the phosphatidylinositol signaling system. Moreover, some of the pathways are directly associated with lipid metabolism; for instance, the phosphatidylinositol signaling pathway enhances lipid metabolism, and insulin secretion inhibits lipolysis. These gene have been associated with various phenotypes including, TG, glucose, ischemic stroke, malaria, BMI, waist and hip circumference Visceral adipose tissue/subcutaneous adipose tissue ratio[27–30].

For fine-mapping, we observed that flashfm produced, on average a smaller CS99 and a larger number of SNPs with MPP > 0.9 than JAM (Fig. 3, Supplementary Data 3). We then compared the CS99 between flashfm and JAM. For HDL, the median credible set size was 24 for the JAM method compared with 18 for flashfm, suggesting a 25% decrease in CS99. For TC, TG and LDL, CS99 decreased by 20.8%, 21.7% and 23.2%,

respectively, suggesting that flashfm enhances the identification and localization of putative genetic variants. Our findings indicate an advantage of leveraging information between traits to jointly fine-map them using flashfm[19]. In addition to refining CS99, compared to JAM single-trait fine-mapping[18], we were able to find noticeably increased MPP for a variant to be causal for a trait, as well as instances where a completely different variant ($r2 < 0.6$) is favoured by flashfm over JAM.

The main strength of this study is that it is the largest GWAS of lipid traits in individuals of African ancestry that covers greater genetic diversity across Africa, while previous studies of African Americans were predominantly of West African ancestry. We used MTAG and flashfm to increase the discovery of additional novel loci and to reduce the credible set size for identifying genetic variants causally associated with lipid traits. The main limitation of this study is the poor generalizability of our findings to individuals of European and Asian ancestry. The GTEx database used for gene expression trait colocalization is predominantly of European ancestry and is not matched for LD with the African ancestry GWAS, which may reduce power. Although our study increased the genetic diversity across Africa, the sample sizes from continental African cohorts were smaller than those of African Americans. Moreover, high heterogeneity was observed among individuals of African ancestry. Nevertheless, we used inverse variance-weighted fixed-effects meta-analysis to address heterogeneity.

In conclusion, by meta-analyzing the summary data from APCDR, GLGC, and AWI-Gen, we identified four novel genomic loci associated with lipid traits in individuals of African ancestry. The multivariate approach identified an additional 14 novel loci, and flashfm, on average, produced a smaller CS99 and the largest number of SNPs with MMP > 0.9. Moreover, our findings highlight the importance of studying African ancestry in the context of GWAS and provide insights into the genetics of dyslipidemia in this population.

## Methods
### Data sources
In the current analysis, we performed a meta-analysis of GWASs of up to 125,000 individuals of African ancestry. Of these individuals ~14,000 were from the African Partnership for Chronic Disease Research (APCDR) consortium in Africa, ~99,000 were from the GLGC and ~11,000 were from the Africa Wits-INDEPTH partnership for Genomic Studies (AWI-Gen). The APCDR comprised the Ugandan General Population Cohort (UGP), the Durban Diabetes Study (DDS), the Durban Case-Control Study (DCC), and the Africa American diabetes mellitus (AADM)[22,31,32]. UGP is a population-based cohort of around 6407 individuals from roughly 25 neighbouring villages of Kyamulibwa, in the countryside of southwest Uganda in East Africa[22]. DDS and DCC are urban population-based cohorts in Durban, South Africa. These cohorts were set up to investigate factors influencing diabetes in South African Zulus in Durban, KwaZulu Natal[31]. The AADM is an ongoing study investigating the genetic epidemiology of non-communicable diseases in Africa. The AADM study comprises 5231 participants recruited from University Medical Centers in Accra and Kumasi in Ghana, Enugu, Ibadan and Lagos in Nigeria and Eldoret in Kenya[32]. The GLGC included data from the Million Veteran Program (MVP AFR), UK Biobank (UKBB, AFR) and other consortia of individuals of African ancestry in the US[16]. The AWI-Gen study participants are drawn from five INDEPTH member centres across the African continent, ensuring a balance of west, east and southern African populations from rural and urban settings. These centres are located in Burkina Faso, Ghana, Kenya and South Africa[33,34].

### Quality control of input summary statistics
Variants with imputation scores lower than 0.3 were removed from all cohorts. Only SNPs with minor allele frequency (MAF) greater than 0.01 in the combined cohort were retained for further analyses. This set of SNPs was then intersected with the 1000 Genomes phase 3

African super population variants (1KAFR) to ensure allelic orientation consistency, estimation of linkage disequilibrium (LD) between test variants and derive estimates of LD scores for downstream analyses[20]. The quality control step of the 1KAFR reference included the removal of duplicated and singleton variants as well as the removal of one individual from the first and second-degree relatives duos. Genomic inflation factors and LD regression intercepts were calculated to assess inflation in univariate summary statistics and correct for residual stratification in the input cohorts. The intercept of the LD scores regression qualifies the amount of inflation in test statistics due to stratification versus polygenicity[35,36]. LD scores within 1 centimorgan (CM) windows and regression intercepts were estimated with the LDSC package.

### Meta-analysis of univariate traits and heterogeneity analyses
We then performed an inverse variance weighted fixed-effect meta-analysis of the GLGC-AFR, APCDR, and AWI-Gen using genome-wide association meta-analysis (GWAMA) for each of the lipid traits[37]. Stratification in the participating cohorts was corrected by inflating standard errors by the square root of the LD score regression intercepts.

### Multi-trait of GWAS for the discovery of loci associated with lipid traits
Multi-trait Analysis of GWAS (MTAG) is a summary statistics-based method for joint analysis that generalizes inverse variance meta-analysis for studies with overlap and different traits[36]. It has been shown to increase the power of association for marginal traits while controlling for confounding of population stratification using LD score regression to estimate confounding bias and trait correlation. Given input summary statistics and LD scores, the algorithm estimates marginal association statistics for each input trait while accounting for other traits. In this step, analysis was restricted to SNPs which were not stranded ambiguous, MAF > 0.01 and with sample size $N > = (2/3)$ of the 90th percentile of variant sample size distribution.

We did not include LDL in our MTAG analysis as it was strongly correlated with other lipid traits ($r2 = 0.92$). Moreover, it is recommended to remove highly correlated traits from an MTAG as it can lead to inflated associations[36], which can result in false positive results. In addition, highly correlated traits can increase the risk of overfitting, which can lead to poor performance and reduce the overall statistical power of the study.

### Single and multi-trait fine-mapping of multiple causal variants
For each of the four lipids traits (HDL, LDL, TC and TG), we identified all genome-wide significant ($p < 5 \times 10^{-8}$) variants, excluding the MHC region due to its extensive LD structure. Independent loci were identified by distance clumping all significant SNPs around lead variants. For fine mapping, initial regions were constructed around each lead variant from each trait, using an interval of ±100 kb. Regions for multi-trait fine-mapping were then constructed by merging regions in which the lead variants had $r2 > 0.5$. This resulted in 47 regions that had genome-wide signals from at least two of the lipid traits, for which we applied both JAM single-trait fine-mapping[18] and flashfm multi-trait fine-mapping[19]. Within these 47 regions, we also assessed whether the traits that did not have a variant with $p < 5 \times 10^{-8}$, had a moderately significant variant with $p < 1 \times 10^{-6}$; signals from such traits were also included in our fine-mapping. The distribution of the number of traits having signals fine-mapped in the same region is as follows: 26 regions with two traits, 14 regions with three traits, and seven regions with four traits (Supplementary Data 4); there are 122 trait-region combinations in total.

We used an in-sample LD reference panel consisting of the Uganda Genome Resource (UGR) and the Zulu cohort, together with the African super-population of 1 K Genomes; first-degree relatives were removed from these cohorts yielding a total of 8850 individuals in the reference panel. Only variants that had a call rate of at least 99%

were retained. flashfm requires the trait correlation matrix for the lipid traits, and we obtained Pearson's correlations based on the in-sample UGR lipids measurements (Fig. 3). Variants with MAF > 0.05 in the GWAS or in the reference panel were included in the fine-mapping, though for five of the dense regions, we used MAF > 0.01; in particular, 5:74556539-74756539, 8:21818089-22018089, 9:107489744-107689744, 9:108081107-108281390, and 19:45300747-45512079. Flashfm gains speed by partitioning the joint Bayes' Factor (BF) into a function of the single-trait BFs and making use of the single-trait fine-mapping results. By default, flashfm considers the SNP models from each trait that have cumulative PP (cpp) of 0.99. We used this default flashfm cpp value of 0.99 for all regions, except for 9:108081107-108281390, and 19:45300747-45512079, where we used cpp=0.95 to further increase computational speed; these two regions had four traits, some with a very large number of models, that noticeably increased the computational speed. Fine-mapping results from JAM and flashfm were each used to generate a 99% credible set (CS99) for each trait flagged in each region. A CS99 was constructed by first sorting all model (multi-SNP) posterior probabilities and consecutively collecting models until the cumulative posterior probability first passes .99; the unique variants from these models form CS99.

For two of our highlighted regions, we display fine-map integrated regional association plots that indicate p-values by location height (y-axis) and MPP by the diameter of the points (Supplementary Fig. 4 and Supplementary Fig. 5); these figures were generated using the web tool flashfm-ivs (http://shiny.mrc-bsu.cam.ac.uk/apps/flashfm-ivis/)[38].

### Gene expression trait colocalization to prioritized genes in the novel loci

In addition to fine mapping, to identify potentially mediating molecular traits in the novel loci identified in the GWAS meta-analysis and MTAG analyses, we performed tissue-specific gene expression quantitative trait loci (eQTL) colocalization analysis for the 18 novel loci. GTEx Version 8 tissue-specific gene expression association summary statistics were downloaded from the eQTL catalogue[39] and merged with meta-analyses and MTAG summary data within the novel loci defined as one MB region around lead SNPs. The merging step involved lifting over GWAS summary data to hg38 build and the removal of duplicated sites. We then estimated evidence of shared causal variants using the coloc.abf function of the *coloc* R package[40]. The coloc.abf estimates the posterior probabilities of 5 hypotheses including a posterior of a shared causal allele between two target traits (H4 PP), two distinct causal alleles (H3 PP), a causal allele for only one target (H1 or H2) and null of no causal allele within a locus. The method was applied using default priors for meta-analysis or MTAGxTissuex-Genexloci combinations and a threshold of H4 PP > 80% was set to assign significance.

### Reporting summary

Further information on research design is available in the Nature Portfolio Reporting Summary linked to this article.

## Data availability

The genome-wide association summary statistics data used in this study are publicly available at https://www.ebi.ac.uk/gwas/downloads/summary-statistics. The summary statistic data generated have been deposited in the GWAS Catalog database under accession codes: GCST90278110 for HDL, GCST90278111 for LDL, GCST90278112 for TG and GCST90278113 for TC. The processed data generated in this study are provided in the Supplementary Information and Supplementary Data.

## Code availability

We used publicly available software GWAMA, MTAG, JAM and flashfm and its code is publicly available at https://bio.tools/

GWAMA, https://github.com/JonJala/mtag, https://github.com/USCbiostats/hJAM, https://github.com/jennasimit/flashfm. Other software programs used are listed and described in the Methods.

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

## Acknowledgements

A.B.K. is supported by the National Institutes of Health/National Human Genome Research Institute (CARDINAL grant 1U01HG011717). S.F. is supported by the Wellcome Trust grant (220740/Z/20/Z). T.C. is an international training fellow supported by the Wellcome Trust grant (214205/Z/18/Z). This work was supported by the UK Medical Research Council (MRC) and the UK Department for International Development (DFID) under the MRC/DFID Concordat agreement, through core funding to the MRC/UVRI and LSHTM Uganda Research Unit. S.M.T. was funded by the West African Center of Excellence for Global Health Bioinformatics Research Training (Award Number: U2RTW010673). S.F. and E.Z. acknowledge the Humboldt-Forschungsstipendium für (AvH) Georg Forster Research Fellowship to S.F. for experienced researchers. J.L.A. and F.Z. are funded by the UK Medical Research Council (MR/R021368/1, MC_UU_00002/4). For the purpose of Open Access, the authors have applied a CC BY public copyright license to any author Accepted Manuscript version arising from this submission.

## Author contributions

S.F. conceptualized the study. S.F., T.C. and J.L.A. designed and supervised the study. A.B.K., S.M.T., F.Z. and O.S. performed the main analyses. A.B.K., S.M.T. and F.Z. wrote the first draft of the manuscript. C.C., M.W., A.B.T., O.N., M.C., M.N., A.C., J.S., S.D., E.Z., and A.P.M. read, reviewed the first draft and provided critical feedback on the paper.

## Competing interests

At the time of writing, M.C. is associated to Cambridge Precision Medicine Limited, UK. However, the remaining authors declare no competing interests.

## Additional information

[1]The African Computational Genomic (TACG) Research Group, MRC/UVRI and LSHTM, Entebbe, Uganda. [2]Malawi Epidemiology and Intervention Research Unit, Lilongwe, Malawi. [3]Sydney Brenner Institute for Molecular Bioscience, Faculty of Health Sciences, University of the Witwatersrand, Johannesburg, South Africa. [4]African Center of Excellence in Bioinformatics, University of Sciences, Techniques and Technologies of Bamako, Bamako, Mali. [5]MRC Biostatistics Unit, University of Cambridge, Cambridge, UK. [6]Faculty of Sciences and Techniques, University of Sciences, Techniques and Technologies of Bamako, Bamako, Mali. [7]H3Africa Bioinformatics Network (H3ABioNet) Node, Center for Genomics Research and Innovation, NABDA/FMST, Abuja, Nigeria. [8]School of Life sciences, University of Westminster, London, UK. [9]Medical Research Council/Uganda Virus Research Institute and London School of Hygiene & Tropical Medicine Uganda Research Unit, Entebbe, Uganda. [10]Department of Biostatistics and Data Science, School of Public Health and Tropical Medicine, Tulane University, New Orleans, LA, USA. [11]Faculty of Medicine and Odonto-stomatology, University of Sciences, Techniques and Technologies of Bamako, Bamako, Mali. [12]Institute of Translational Genomics, Helmholtz Zentrum München - German Research Center for Environmental Health, Neuherberg, Germany. [13]TUM School of Medicine, Translational Genomics, Technical University of Munich and Klinikum Rechts der Isar, Munich, Germany. [14]Centre for Genetics and Genomics Versus Arthritis, Centre for Musculoskeletal Research, University of Manchester, Manchester, UK. [15]Channing Division of Network Medicine, Brigham and Women's Hospital, Boston, MA, USA. [16]Harvard Medical School, Boston, MA, USA. [17]MRC/Wits Developmental Pathways for Health Research Unit, Department of Pediatrics, Faculty of Health Sciences, University of the Witwatersrand, Johannesburg, South Africa. [18]Department of Non-Communicable Disease Epidemiology, London School of Hygiene and Tropical Medicine, London, UK. [19]These authors contributed equally: Abram Bunya Kamiza, Sounkou M. Touré, Feng Zhou. ✉e-mail: segun.fatumo@lshtm.ac.uk

