## [Peer Review File · Nature Communications]

Multi-trait discovery and fine-mapping of lipid loci in 125,000 individuals of African ancestryREVIEWER COMMENTS

Reviewer #1 (Remarks to the Author):

The study of AB Kamiza et al, aimed at identifying novel genetic variants associated with lipid traits in populations of African ancestry including up to 125,000 individuals, using multi-trait analysis of genome-wide association studies (MTAG), and multi-trait fine mapping, to boost the statistical power of detecting novel genetic loci, that are causally associated with dyslipidemia.

By adding 14k individuals from the African Partnership for Chronic Disease Research (APCDR) consortium in Africa, and 11k from Africa Wits-INDEPTH Partnership for Genomic, to the 99k individuals included in the last GLGC analysis, they have been able to identify 14 novel genetic loci compared to the last findings from GLGC in this ethnic group, and some of them seems to be causal for dyslipidemia.

This is an important effort given the lack of evidence on lipid metabolism pathways and their links to CVD in populations of African ancestry. Although I do not have concerns on the methods used to conduct the GWAS meta-analysis, and the MTAG, I have some major concerns on the lack of genomic functional results and the interpretation/discussion of the results:

- In the method section, they mentioned having performed gene expression trait colocalization, and functional annotation, enrichment analysis using eQTL for the novel loci they identified, but these results have not been reported. Given the identification of novel loci, post-GWAS functional analysis is a key step to validate the GWAS findings and understand through which pathways they might contribute to dyslipidemia, and whether novel SNPs causally associated with dyslipidemia might also be causal for CVD.

- The discussion is very succinct and does not follow a logical flow with the presented results. There is no discussion on findings from the meta-analysis (eg MGAT2 is known to be involved in synthesis of TG in the liver). They discussed findings on MGST1, PLPPR1, NFATM1, but these genes are not reported in the result section. There is no discussion of the limitations of the study.

Reviewer #2 (Remarks to the Author):

Overall, I think that this is a nice approach to important work on underrepresented individuals. A few recommendations for improvement/clarification:

Given the focus on multi-trait approaches, I think that more of the motivation for this study should be on why statistically and biologically the authors expect this to be an improvement on single-trait methods.

I would restate the first sentence of the abstract. Most GWAS of lipids focus on multiple lipids, but I think that the intent of the authors was that most focus on a single lipid at a time.

In the abstract (and perhaps throughout) I would describe differences in terms of a single- vs. multi-trait approach instead of listing the software tools, which may be unfamiliar and are less relevant for understanding the strategy.

Should avoid use of the term "racial".

I would suggest adding "multi-trait" to the title, as much of the manuscript is devoted to showing the improvement by using multi-trait approaches.

This is not a study of dyslipidemia. The authors use this term repeatedly through the paper, but a study of dyslipidemia would require running analyses on binary traits defined by thresholds of lipid values (high TG, high LDL, low HDL). The described analyses are all on the quantitative traits.

Ln 69-71: This is a rather old and weak reference for this point, which is unnecessary given that this is a topic that is very well described in the literature. The authors could consider reviewing research from RODAM, which looks at the distribution of cardiometabolic traits in individuals of different ancestries living in the same location. Or, if you would like to report on prevalence of cardiometabolic disorders in the US, perhaps the Center for Disease Control's National Health Statistics or National Vital Statistics reports. Also, data from NHANES is designed to be nationally representative of the US population, and there are many publications giving cardiometabolic disease distribution.

Ln 72, "Dietary intake" is too broad a term to be included in this list as being associated with an atherogenic lipid profile. A healthy dietary intake, obviously, is not positively associated with this outcome

Ln 84-86: I think a more relevant point here is that African Americans do not carry ancestry from across the continent of Africa. They have predominantly West African ancestry. For this reason (and, obviously, the different environmental background), African Americans cannot be expected to be suitable proxies for all Africans.

Ln 116: What happened with LDL? Were there no associations? Was LDL omitted from the MTAG analyses for some reason?

Ln 177: You mean here that their published findings do not show the SNP ids because there are >10 variants with $MPP > 0.9$? I would at least attempt to contact the authors to see if they would be willing to help you find the information you are looking for. This is a bit unsatisfying as a reason to not compare findings.

Ln 246: Please make it clear what data included are primary and what is secondary use of summary statistics. I believe that many of these are secondary use of summary statistics. In that case, please state this and provide details on where these data are accessible. The "genotyping and quality control of raw data" is particularly confusing on this point, since I don't think that processing the raw genotypes was something that was done for all of the included studies.

Ln 286: Here it is mentioned that only $MAF > 0.01$ was retained for analysis, but different (lower) MAF thresholds are listed in other sections of the paper (Ln 309-310 $MAF > 0.001$; Ln 329 $MAF > 0.005$).

Ln 298: The authors list a meta-analysis of "APCDR, MVP-AFR, and GLCC-AFR" – I assume that the latter is GLGC-AFR. Was AWI-GEN not included here? Also, was the data from APCDR included together in one meta-analysis? I mean was it a fixed effects meta of APCDR + MVP-AFR + GLGC-AFR or was it UGP + DDS + DCC + AADM + MVP-AFR + GLGC-AFR? If the former, that seems a little problematic to put East, South, and West Africans all as a single unit of analysis. I would expect quite a bit of heterogeneity between these groups.

Relatedly, I think it is a bit of a missed opportunity not to examine similarities/differences across the different African ancestries included. For instance, it would be wonderful to see a forest plot of some of the main hits across African (and non-AFR) ancestries. Were these all consistent?

Please provide an explanation for the exclusion of the MHC region.

Ln 314: Which reference was used for this LD pruning?

Ln 334: The computation speed made it impractical to use the same settings as for the other regions?

Table 2: This would be easier to read with horizontal lines or shading to indicate different loci. Also, these seem to be sorted by Chromosome and position for the most part, but there are 2 entries for

rs10819792 that are not sequential.

Ln 548: "Venn" diagram

Reviewer #1

The study of AB Kamiza et al, aimed at identifying novel genetic variants associated with lipid traits in populations of African ancestry including up to 125,000 individuals, using multi-trait analysis of genome-wide association studies (MTAG), and multi-trait fine mapping, to boost the statistical power of detecting novel genetic loci, that are causally associated with dyslipidemia.

By adding 14k individuals from the African Partnership for Chronic Disease Research (APCDR) consortium in Africa, and 11k from Africa Wits-INDEPTH Partnership for Genomic, to the 99k individuals included in the last GLGC analysis, they have been able to identify 14 novel genetic loci compared to the last findings from GLGC in this ethnic group, and some of them seems to be causal for dyslipidemia.

This is an important effort given the lack of evidence on lipid metabolism pathways and their links to CVD in populations of African ancestry. Although I do not have concerns on the methods used to conduct the GWAS meta-analysis, and the MTAG, I have some major concerns on the lack of genomic functional results and the interpretation/discussion of the results:

Response: Thank for you your thoughtful comments and suggestion. In this revision, we have included the genomic functional results, interpretation and discussion of the findings. These additional changes are highlighted in red and tracked changes.

- In the method section, they mentioned having performed gene expression trait colocalization, and functional annotation, enrichment analysis using eQTL for the novel loci they identified, but these results have not been reported. Given the identification of novel loci, post-GWAS functional analysis is a key step to validate the GWAS findings and understand through which pathways they might contribute to dyslipidemia, and whether novel SNPs causally associated with dyslipidemia might also be causal for CVD.

Response: Thank you for your comments and suggestion. In this revision, we have included the results of gene expression trait colocalization, and functional annotation on lines 197 -209 on page 8 and 9. "To identify putative functional mechanisms of novel loci, we performed Bayesian colocalization on meta-analysis and MTAG summary data with GTEx v8 tissue-specific gene expression quantitative trait loci (eQTL) using coloc 22. Of the four novel loci identified by meta-analysis, none were colocalized (H4 PP <0.8) with the respective lipid traits (Table.S6). For the MTAG summary data, we found six genetic loci (rs13438114, rs6477710, rs113951466, rs4743855, rs10819792, and rs7041800) with H4 PP > 0.8; the co-localized gene expression traits involved multiple tissues (Table.S7). For instance, ADCYAP1R1, ANKRD19P, CCDC180, ECM2, FGD3, HABP4, LINC02937, MSANTD3, NUTM2G, OGN, IPPK, PTLC1, and ZNF484 had H4 PP > 85% with MTAG HDL and were expressed in several tissues, notably those of the liver, brain, adipose tissues, artery, gastrointestinal tract, and whole blood (Table.S7). Interestingly, these genes are involved in several pathways, including insulin secretion, renin secretion, the cAMP signaling pathway, inositol phosphate metabolism, metabolic pathways, and the phosphatidylinositol signaling system"

- The discussion is very succinct and does not follow a logical flow with the presented results. There is no discussion on findings from the meta-analysis (eg MGAT2 is known to be involved in synthesis of TG in the liver). They discussed findings on MGST1, PLPPR1, NFATM1, but these genes are not reported in the result section. There is no discussion of the limitations of the study.

Response: Thank you for your comment and suggestion. We have further discussed the gene reported in Table 1 and Table 2 of this study on lines 222- 232 “ Our meta-analysis of AWI-Gen, APCDR, and GLGC-AFR, found four novel loci associated with lipid traits in individuals of African ancestry (Table 1). The rs115505361 is mapped on MGAT2 and associated with body mass index, blood proteins, insulin sensitivity and hypertriglyceridemia. MGTA2 is involved in various metabolic and N-glycan biosynthesis pathways, including lipid metabolism and protein synthesis. rs12118522, mapped to CHRM3, is associated with hypertension and body fat distribution. Some pathways associated with CHRM3 include calcium signalling, cholinergic synapses, taste transduction, insulin secretion, salivary secretion, gastric acid secretion, and pancreatic secretion pathways. rs2451303 and rs6704760 are novel loci associated with lipid traits and are mapped in the intergenic regions between NT51B and RDH14, MYCN and MYCNOS, respectively. These genes have been previously reported to be associated with cardiovascular diseases and neoplasm, respectively.”

Further discussion are on lines 240-255 “ Notably, our novel genetic loci have been linked to traits correlated with blood lipid traits, including body mass index, systolic and diastolic blood pressure, diabetes, chronic kidney disease, blood protein levels, blood and immunology phenotypes, and Alzheimer’s biomarkers such as amyloid or infectious diseases. For instance, CACNA1E encodes the alpha-1E subunit of R-type calcium channels, which belong to the 'high-voltage activated' group that may be involved in the modulation of firing patterns of neurons important for information processing 23. This gene is involved in the following pathways; MAPK signaling, calcium, T2D, and early infantile epileptic encephalopathy pathways and has been previously associated with blood pressure, BMI, and coronary artery calcification. TMEM64 is localized in the endoplasmic reticulum and modulates the nuclear localization of β -catenin, resulting in the activation of β -catenin-mediated transcription 24. Diseases and phenotypes associated with TMEM64 include cholesterol, creatinine, chronotype, and serum metabolite concentrations in chronic kidney disease. PPP2R2C has previously been associated with BMI, coronary artery disease, T2D, and metabolite levels. This gene belongs to the phosphatase 2 regulatory subunit B family 25, which is related to PI3K-Akt, AMPK signaling, adrenergic signaling in cardiomyocytes, Chagas disease, hepatitis C, and human papillomavirus infection pathways“

We have also discussed the strength and the limitation of the study in lines 278 -289 “ The main strength of this study is that it is the largest GWAS of lipid traits in individuals of African ancestry that covers greater genetic diversity across Africa, while previous studies of African Americans were predominantly of West African ancestry. We used MTAG and flashfm to increase the discovery of additional novel loci and to reduce the credible set size for identifying genetic variants causally associated with lipid traits. The main limitation of this study is the poor generalizability of our findings to individuals of European and Asian ancestry. The GTEx database used for gene expression trait colocalization is predominantly of European ancestry and is not matched for LD with the African ancestry GWAS, which may reduce power. Although our study increased the genetic diversity across Africa, the sample sizes from continental African cohorts were smaller than those of African Americans. Moreover, high heterogeneity was observed among individuals of African ancestry. Nevertheless, we used inverse variance-weighted fixed-effects meta-analysis to address heterogeneity”

.

“

Reviewer #2

Overall, I think that this is a nice approach to important work on underrepresented individuals. A few recommendations for improvement/clarification:

Given the focus on multi-trait approaches, I think that more of the motivation for this study should be on why statistically and biologically the authors expect this to be an improvement on single-trait methods.

Response: Thank you for your comments and suggestion. The motivation for using a multi-trait over the single trait is that the multi-trait approach is an improvement of single-trait methods for several reasons. Firstly, by analysing multiple lipid traits together, we accounted for the correlations between the traits, which lead to more accurate and robust results than a single-trait method. Moreover, single-trait methods may not capture some genetic variants that affect multiple lipid traits simultaneously, and their true effects may not be fully known. Secondly, a multi-trait approach can provide insights into shared biological pathways that underlie different traits. By identifying genetic variants that affect multiple traits, we can gain a better understanding of the biological mechanisms that govern the association between different traits. Lastly, by analysing multiple lipid traits together, we minimise the issue of multiple testing, which can arise when analysing a large number of traits separately.

I would restate the first sentence of the abstract. Most GWAS of lipids focus on multiple lipids, but I think that the intent of the authors was that most focus on a single lipid at a time.

Response: Thank you for your comment and suggestion, I this revision we have restated the first sentence of the abstract to “Most of the genome-wide association studies (GWAS) for lipid traits focus on analysing lipid traits separately, one at a time “

In the abstract (and perhaps throughout) I would describe differences in terms of a single- vs. multi-trait approach instead of listing the software tools, which may be unfamiliar and are less relevant for understanding the strategy.

Response: Thank you for your thoughtful comment and suggestion. In this revision, we have stated the differences in the single vs multi-trait approach on lines 52-53 and 96-97 of this manuscript. “Analyzing multiple lipid traits simultaneously can provide more accurate and robust results than analyzing each trait separately “ and lines 98-101 “Moreover, we used a single trait (JAM) [18] and multi-trait (flashfm) [19] fine-mapping methods to identify causal genetic variants associated with lipid traits in individuals of African ancestry; sharing information between traits by joint fine-mapping with flashfm results in higher resolution (smaller credible sets) than fine-mapping each trait independently.”

Should avoid use of the term “racial”.

Response: Thank you for your comment and suggestion, in this revision, we have replaced the word “racial” with “ancestry”.

I would suggest adding “multi-trait” to the title, as much of the manuscript is devoted to showing the improvement by using multi-trait approaches.

Response: Thank you for your thoughtful suggestion. In this revision, we have included multi-trait in our title. Now the title of the manuscript is “Multi-trait discovery and fine-mapping of lipid loci in 125,000 individuals of African ancestry”

This is not a study of dyslipidemia. The authors use this term repeatedly through the paper, but a study of dyslipidemia would require running analyses on binary traits defined by thresholds of lipid values (high TG, high LDL, low HDL). The described analyses are all on the quantitative traits.

Response: Thank you for your thoughtful comment and suggestion, we have revised and replaced the term “dyslipidaemia” with lipid traits in this revision. These changes can be found throughout the manuscript.

Ln 69-71: This is a rather old and weak reference for this point, which is unnecessary given that this is a topic that is very well described in the literature. The authors could consider reviewing research from RODAM, which looks at the distribution of cardiometabolic traits in individuals of different ancestries living in the same location. Or, if you would like to report on prevalence of cardiometabolic disorders in the US, perhaps the Center for Disease Control’s National Health Statistics or National Vital Statistics reports. Also, data from NHANES is designed to be nationally representative of the US population, and there are many publications giving cardiometabolic disease distribution.

Response: Thank you for your comment and suggestion. In this revision, we have updated our references and cited the national representative studies see lines 383- 391 of this manuscript.

1. Linden, E. van der et al. Dyslipidaemia among Ghanaian migrants in three European countries and their compatriots in rural and urban Ghana: The RODAM study. *Atherosclerosis* 284, 83–91 (2019).
2. Kanchi, R. et al. Gender and Race Disparities in Cardiovascular Disease Risk Factors among New York City Adults: New York City Health and Nutrition Examination Survey (NYC HANES) 2013-2014. *J Urban Health* 95, 801–812 (2018).
3. van der Linden, E. L. et al. The prevalence of metabolic syndrome among Ghanaian migrants and their homeland counterparts: the Research on Obesity and type 2 Diabetes among African Migrants (RODAM) study. *Eur J Public Health* 29, 906–913 (2019).

Ln 72, “Dietary intake” is too broad a term to be included in this list as being associated with an atherogenic lipid profile. A healthy dietary intake, obviously, is not positively associated with this outcome

Response: Thank you for your suggestion. In this revision, we have indicated that unhealthy dietary intake, physical inactivity, cigarette smoking, alcohol consumption and several genetic factors are some of the factors implicated in atherogenic lipid levels. These changes are on lines 73-75 of this manuscript.

Ln 84-86: I think a more relevant point here is that African Americans do not carry ancestry from across the continent of Africa. They have predominantly West African ancestry. For this reason (and, obviously, the different environmental background), African Americans cannot be expected to be suitable proxies for all Africans.

Response: Thank you for your thoughtful comments and suggestions in this revision, we have indicated that although the GLGC meta-analysis included 99,432 individuals of African ancestry, 72,859 were African Americans in the US, which may not carry ancestry-specific genetic variants across Africa. Moreover, the majority of African Americans in the US carry West African ancestry. To further identify genetic loci associated with lipid traits in individuals of African ancestry and determine its molecular mechanisms, putative causal genetic variants and improve diversity we performed a meta-analysis of GWAS including up to 125,000 individuals of African ancestry. These changes are on lines 85-91 of this manuscript.

Ln 116: What happened with LDL? Were there no associations? Was LDL omitted from the MTAG analyses for some reason?

Response: Yes, we did not include LDL in MTAG analysis as it was strongly correlated with other lipid traits ($r^2=0.92$). Moreover, it is recommended to remove highly correlated traits from an MTAG as it can lead to inflated associations [1], which can result in false positive results. In addition, highly correlated traits can increase the risk of overfitting, which can lead to poor performance and reduce the overall statistical power of the study. We have noted this in the Methods section on MTAG.

1. Turley, P., Walters, R. K., Maghzian, O., Okbay, A., Lee, J. J., Fontana, M. A., ... & Davis, J. M. (2018). Multi-trait analysis of genome-wide association summary statistics using MTAG. *Nature genetics*, 50(2), 229-237.

Ln 177: You mean here that their published findings do not show the SNP ids because there are >10 variants with $MPP>0.9$? I would at least attempt to contact the authors to see if they would be willing to help you find the information you are looking for. This is a bit unsatisfying as a reason to not compare findings.

Response: Thank you thoughtful comment. In our original comparisons we had used the GLGC credible set variants, which lists all variants in the 99% credible set, if there are not more than 10 variants. We have now made a fairer comparison using the available information provided in the GLGC supplementary material, comparing against the variants with $PP>0.90$ in the GLGC analysis. The updated text is given below and Supplementary Table 4 has been appropriately updated. We have also included an additional table (Supplementary Table 5) that gives the details of all of our prioritised variants, including all functional annotations (not only most severe) and whether they were also prioritised ($PP>0.90$) in the GLGC analyses.

"Among these 62 regions, 31 have variants with $PP>0.90$ in the GLGC African analysis, and 48 have variants with $PP>0.90$ in the GLGC trans-ancestry analysis; we compare our results with those of GLGC within these regions where both flashfm and one of the GLGC analyses has at least one variant with $MPP > 0.90$.

As expected, there is a higher agreement between our analysis with the African analysis of GLGC, than with their trans-ancestry analysis. In 45% (14/31) of the regions that have variants prioritised by both flashfm and the African fine-mapping of GLGC, there was a shared prioritised variant; 21% (10/48) of the regions that have variants prioritised by both flashfm and the multi-ancestry fine-mapping of GLGC share a prioritised variant."

Ln 246: Please make it clear what data included are primary and what is secondary use of summary statistics. I believe that many of these are secondary use of summary statistics. In that case, please state this and provide details on where these data are accessible. The "genotyping and quality control of raw data" is particularly confusing on this point, since I don't think that processing the raw genotypes was something that was done for all of the included studies.

Response: Thank you for your thoughtful comment and suggestions. In this manuscript, we used GWAS summary data from individuals of African ancestry in GLGC, APCDR, and AWI-Gen. These summary data were obtained from online public repository. Moreover, in this revision, we have removed the section called "genotyping and quality control of raw data" in the methods section of the manuscript on page 11.

Ln 286: Here it is mentioned that only $MAF > 0.01$ was retained for analysis, but different (lower) MAF thresholds are listed in other sections of the paper (Ln 309-310 $MAF > 0.001$; Ln 329 $MAF > 0.005$).

Response: Thank you for your comment. We have corrected the typo on line 309-310 and line 329. Now, the MAF on line 309-310 is 1% and on line, 329-310 is 5%.

Ln 298: The authors list a meta-analysis of “APCDR, MVP-AFR, and GLCC-AFR” – I assume that the latter is GLGC-AFR. Was AWI-GEN not included here? Also, was the data from APCDR included together in one meta-analysis? I mean was it a fixed effects meta of APCDR + MVP-AFR + GLGC-AFR or was it UGP + DDS + DCC + AADM + MVP-AFR + GLGC-AFR? If the former, that seems a little problematic to put East, South, and West Africans all as a single unit of analysis. I would expect quite a bit of heterogeneity between these groups.

Response: Thank you for your comment. Yes we included AWI-Gen summary data in the analysis see line 89- 91 “Of these individuals ~14,000 were from the African Partnership for Chronic Disease Research (APCDR) consortium in Africa, ~99,000 were from the GLGC and ~ 11,000 were from Africa Wits-INDEPTH Partnership for Genomic Research (AWI-Gen) in Africa”. In this analysis, we used three summary statistic dataset namely GLGC-AFR, APDCR and AWI-Gen. For GLGC_AFR the following cohort were used in their meta-analysis ARIC, BioMe, CARDIA, CFS, CHOP, CHS, eMERGE, HABC, HANDLS, HRS, HyperGen_AA, HyperGen- Axiom, JHS, JoCoOA, Jupiter, MESA, MVP, PMBB, UKB, and WHI-AA using fixed METAL. For APDR, UGR, AADM, DDS and DCC were meta-analysed using Han-Eskin random effects meta-analysis implemented in METASOFT. For AWI-Gen, Nanoro (west Africa), Navrongo (west Africa), Dikgale (south Africa), Agincourt (south Africa), SOWETO (south Africa), and Nairobi (east Africa) cohort were meta-analysis using Han and Eskin’s Random Effects model (RE2) implemented in METASOFT to address the heterogeneity from different geographical region. In our analysis, we performed an inverse variance weighted fixed-effect meta-analysis of the GLGC-AFR, APCDR, and AWI-Gen using genome-wide association meta-analysis (GWAMA) for each of the lipid traits see line 298-300 of this manuscript.

Relatedly, I think it is a bit of a missed opportunity not to examine similarities/differences across the different African ancestries included. For instance, it would be wonderful to see a forest plot of some of the main hits across African (and non-AFR) ancestries. Were these all consistent?

Response: Thank you for your thought comment and suggestion. We totally agree with you on examining the similarities and differences among Africa cohort included. However, we do not have access to individual cohort data. What we have are summary data from GLG_AFR, APCDR and AWI-Gen. These are meta-analysis summary data from small cohort within the respective cohort. For instance, APCDR summary data included Uganda genome resource, Durban Diabetes Study, Durban Case-Control and African American Diabetes Mellitus study. See the comment above. Nevertheless, we have compared the similarities and differences among GLGC-AFR, APCDR and AWI-Gen on lines 116-119 “ To determine the similarities or differences across the three datasets used in the meta-analysis, we identified the top hits in GLGC-AFR compared to those in APCDR and AWI-Gen (Fig.S1). We found that these loci were consistent across the three datasets, although some were not genome-wide significant in the APCDR and AWI-Gen datasets “

Please provide an explanation for the exclusion of the MHC region.

Response: Thank you for the comment. We excluded the MHC region due to its extensive LD structure, see lines 335-336 “For each of the four lipids traits (HDL, LDL, TC and TG), we

identified all genome-wide significant ($p < 5 \times 10^{-8}$) variants, excluding the MHC region due to its extensive LD structure “

Ln 314: Which reference was used for this LD pruning?

Response: Thank you for your comments. Now on line 352 of the revision we have stated that “Independent loci were identified by distance clumping all significant SNPs around the lead variants”

Ln 334: The computation speed made it impractical to use the same settings as for the other regions?

Response: Thank you for your thoughtful comments. Flashfm gains speed by partitioning the joint Bayes' Factor (BF) into a function of the single-trait BFs and making use of the single-trait fine-mapping results. By default, flashfm considers the SNP models from each trait that have cumulative PP of 0.99. For two regions, we decreased this cpp to 0.95 because these regions had four traits, some with a very large number of models that noticeably increased the computational speed, lines 356-362 “Flashfm gains speed by partitioning the joint Bayes' Factor (BF) into a function of the single-trait BFs and making use of the single-trait fine-mapping results. By default, flashfm considers the SNP models from each trait that have cumulative PP (cpp) of 0.99. We used this default flashfm cpp value of 0.99 for all regions, except for 9:108081107-108281390, and 19:45300747-45512079, where we used cpp=0.95 to further increase computational speed; these two regions had four traits, some with a very large number of models, that noticeably increased the computational speed “

Table 2: This would be easier to read with horizontal lines or shading to indicate different loci. Also, these seem to be sorted by Chromosome and position for the most part, but there are 2 entries for rs10819792 that are not sequential.

Response: Thank you for your comment and suggestions. In this revision, we have shaded Table 2 to indicate different genetic loci. Yes, in Table 2, we have loci that are associated with different lipid traits and have different effect sizes; hence, we presented them multiple times depending on the number of lipid traits it is associated with.

Ln 548: “Venn” diagram

Response: Thank you for the suggestion; we have corrected the word “Venni” to Venn diagram on line 548 of the manuscript.

REVIEWERS' COMMENTS

Reviewer #1 (Remarks to the Author):

The authors have notably improved their manuscript, and have addressed my comments appropriately. I do not have any further suggestions.

Reviewer #2 (Remarks to the Author):

I am satisfied with the edits that have been made to this manuscript and find it suitable for publication.